# The Invisible Discrimination: Biases in the Clinical Approach Regarding Migrants: A Study to Help Ethnopsychology Services and Clinicians

**DOI:** 10.3390/bs14030155

**Published:** 2024-02-21

**Authors:** Antonio Iudici, Lucia Colombo, Simona Carla Silvia Caravita, Paolo Cottone, Jessica Neri

**Affiliations:** 1Department of Philosophy, Sociology, Education and Applied Psychology, University of Padova, 35122 Padova, Italy; paolo.cottone@unipd.it (P.C.); jessica.neri@phd.unipd.it (J.N.); 2Institute of Psychology and Psychotherapy (Scuola Interazionista), 35100 Padua, Italy; scuola@scuolainterazionista.it; 3Norwegian Centre for Learning Environment and Behavioral Research in Education, University of Stavanger, N-4036 Stavanger, Norway; simona.c.caravita@uis.no

**Keywords:** bias, immigrants, clinical psychology, health care, public service, health promotion

## Abstract

The complexity of migration flows across the world has led to a redefinition of psychological and social services users. The access of migrants from different cultural backgrounds to clinical services or social health services has diversified the demand for concomitant help. Biases and misinterpretations have been created by unaccustomed professionals in this field, which could lead to serious consequences and invalidate diagnostic and treatment procedures. The purpose of this study is to summarize the evidence about errors or prejudices observed in clinical practices regarding the provision of social health services to people from different cultural backgrounds. Results show three main types of biases: racial stereotype activation, ethnocentrism and micro-aggressions. Some implications on the clinical setting were discussed, as being aware of these biases can help mental health professionals manage communication more consciously with users.

## 1. Introduction

Despite the fact that the history of humanity has been characterized by innumerable migrations over time, in the last several years, globalization has contributed to the creation of a network that necessitates more intensive interactions between individuals and communities with regard to migration [1,2]. More immediately, recent wars in Syria, Afghanistan, and Ukraine are leading to an increase in the number of refugees across the world who are likely to seek psychological (and not only material) help.

In light of these phenomena, everyday life experiences in the current world are becoming richer and more diversified [3,4,5]. There is an urgent need to develop procedures of interpretation to read and understand these new processes. Moreover, the increasing number of second- and third-generation immigrants worldwide renders critical the definition of identity constructed only on the basis of ethnicity. Words such as “immigrant”, “foreigner” and “culture” are no longer sufficient to capture the complexity of this process, which is produced by the continuous increase in migratory flows. Specifically, the use of these labels to refer to a wide range of situations, together with the human cognitive need to process the unknown as if it were already known [6,7,8,9], can have serious consequences in the context of clinical services provided to migrants. In the abovementioned context, the access of migrants or culturally different individuals to public health services is an issue that has provoked an increasing number of studies concerning cultural differences and the development of intervention programs ad hoc [10]. Since the 1970s, scholars have focused on the impact that patient variables such as race can have on mental health professionals’ clinical judgments, often adopting an archival method to investigate epidemiologic results about diagnosis rate [11,12,13] and type of treatment [14,15] among patients deemed as having a different cultural background. These studies aimed to clarify the influence of the variable named “culture” within the mental health field, thus initiating the emergence of ethnopsychology [16]. Since then, many studies have investigated how culture may impact the role and work of mental health professionals and patients’ concomitant responses. First, some authors have highlighted a clear difference in identifying mental health symptoms according to the cultural background of a patient [17]. Other scholars have noted a lack of validity of related tests and assessment instruments [18,19], some errors of judgment made by mental health professionals [20,21,22] and bias based on bullying [23]. Some researchers have also investigated the tendency of clinicians and their services to interpret behaviors specific to certain cultures using the diagnostic symptoms of Western psychiatry [24], which amounts to the implementation of a form of ethnocentrism [25]. With regard to patients, some authors have highlighted that poor use and certain inequalities in public mental health services [26,27] added to a clearly identifiable distrust toward Western therapy [28]. All these can have a direct impact on the well-being of many people who come to clinicians. Moreover, as discussed above, in many countries, migration flows have raised new issues involving services and clinicians, who need a more precise and error-free key that will help them understand their responsibilities. In fact, these social changes may also affect the way clinicians usually operate [29] or the kinds of training they need to avoid compromising their operational work in shifting circumstances [30]. It is essential to focus on the necessary skills and knowledge that clinicians must acquire to avoid the aforementioned errors. In 2017, the American Psychological Association [31] (p. 7) published a set of guidelines meant to “provide psychologists with a framework from which to consider evolving parameters for the provision of multiculturally competent services”. In recent years, many studies have focused on clarifying the role of racial stereotypes and biases in the medical field [32,33,34]. According to these studies, the variable of culture can indeed affect a medical diagnosis; thus, the necessity to explore cultural processes in the field of psychological services arises. Crucially, the clinical encounter can be affected by difficulties and criticalities that result in the adoption of cognitive shortcuts, which may, in turn, lead to assessment errors [35]. The use of these shortcuts is usually aimed at simplifying the clinical relationship between patients and their mental health professionals; however, they can lead to biases. Our study is, therefore, guided by the following research questions: In the literature, what are the main biases that can be detected among mental health professionals when they are confronted with people from different cultural backgrounds? How do the relevant studies describe the implications of these biases? Thus, our systematic review aims to summarize the main biases identified in the literature as occurring during the clinical encounter between migrants or culturally different patients and mental health professionals. Our study adopts an interactionist epistemological framework [36,37]. According to this position, reality is not given, but it is a process that develops via the process of communication between individuals (interactions). This position differs from the mechanistic (or ontological) paradigm, which postulates reality as external to the observer, governed by empirical laws within ontological entities, such as causality. Sure enough, when we refer to “culture”, we are not discussing a defined and static entity but something adaptive and changeable precisely because it is the result of social negotiation [38]. 

## 2. Method

To answer our research questions, we conducted a systematic review. Our methodology was chosen for its strengths. According to Grant and Booth [39], the systematic review method seeks to systematically search for, appraise and synthesize research evidence, often adhering to guidelines on the conduct of a review. Unfortunately, pertinent studies in the intercultural field lack methodological rigor in terms of the search strategy of materials. For this reason, our study adopted a methodology based on a rigorous analysis and an accurate evaluation of the available literature. 

Our primary objective was to provide the reader with an overview of the so-called “primary sources” [40], summarizing the main biases in the literature identified as occurring during the clinical encounters between migrants or culturally different patients and mental health professionals. 

On our chosen topic, there are many studies that have not been formalized in scientific publications. Moreover, related public opinion has produced a vast number of online posts and communications. For this reason, we chose to conduct a systematic review of this topic in order to provide an overview of the main scientific results obtained regarding our study topic [41,42]. In turn, we found a considerable increase in the amount of research in this area over time; we intend to provide reliable accounts of the extant research in this review [43]. We systematized and summarized the available data using a systematic review method and in accordance with the PRISMA (Preferred reporting items for systematic review and meta-analysis protocols) protocol. 

### 2.1. Inclusion and Exclusion Criteria

The inclusion criteria were the following: (1) full-length articles published in peer-reviewed journals in the English language from 1980 to 2022; (2) empirical articles (quantitative or qualitative) related to the biases shown by professionals toward/against patients with different cultural backgrounds. The following types of articles were excluded: (1) articles that were not peer-reviewed, such as theses, book chapters and conference papers; (2) articles addressing different topics or not specifically including biases regarding patients with different cultural backgrounds; (3) studies on rural-to-urban migrant workers. 

### 2.2. Information Sources and Search Strategy

To collect data concerning our aim, we used Scopus as a database. This was because we found Scopus to be the largest and most thorough international database of peer-reviewed literature, including publications offered by other databases. Compared with other databases, Scopus seemed more pertinent to our review. The search strategy was executed by combining keywords pertaining to three different subject areas:-Mental health: psychology, psychotherapy, psychiatry, counseling;-Culture and cultural background: cultural, culture, multicultural, minority group, racial, ethnic, race, racism;-Bias and judgments errors: bias, mistake, stereotyping, prejudice.

Regarding the use of different keywords, our aim was to reach sources discussing the same issue by referring to it differently. We were aware that errors, bias and mistakes are words with different meanings. Nevertheless, since defining them theoretically is beyond the purpose of our study, in this article, we use them as synonymous words.

### 2.3. Data Screening and Extraction

We excluded articles written before 1980. After an initial analysis, we included the word “clinical” in each keyword set since most documents addressed the issue in a non-clinical manner. In turn, we found 629 articles. Afterward, we added the following search strings: “AND NOT (family or couple or marriage)” and “AND NOT (kids or adolescent)” to exclude family therapy, couple’s therapy and therapy with children. The searches addressing these areas were, in fact, beyond the scope of this study. The 377 documents we subsequently selected also excluded literature reviews and books (secondary sources), as we limited the search according to the “document type” of “article”. Thereafter, the total number of remaining articles was 316. These were reviewed in terms of “title” and “abstract” so we could exclude the non-pertinent articles. More precisely, our exclusion criteria concerned the type of treatment (e.g., group psychotherapy) and the research area (e.g., assessment instruments and usage of mental health public services). After excluding 283 articles, the final number of the selected documents became 33. Details of this process are provided below and displayed in the flow chart in Figure 1. The 33 articles covered a period of 39 years, from 1980 to 2022. They were equally divided among the three main outcomes. All the articles we found were written in English, and the majority (99.9%) of the studies were conducted in the United States of America (USA). All the selected articles are displayed in Table 1.

**Prisma Statement:** 
*The reporting of this systematic review was guided by the standards of the preferred reporting items for systematic review and meta-analysis (PRISMA) statement.*


It is essential to mention that most of the articles we selected examined the study topic using two methodologies. One methodology involved presenting a clinical study in a video, audio, or textual format to several participants (psychologists, therapists, counselors or psychiatrists). Subsequently, the participants answered a questionnaire about global evaluation, clinical judgments, their severity and the perception of factors that can emerge during therapy.

The other modality involved analyzing the epidemiological characteristics of a specific sample in order to figure out the reason why mental health professionals choose one diagnosis over another. These two modalities were aimed at investigating if and how culture is a variable that can influence the clinical judgment or the perception of a case’s severity. Only two articles addressed the above issue using a different methodology. One of them [56] was a phenomenological inquiry conducted by distributing questionnaires to 108 members of a multicultural counseling and psychotherapy training organization; the questionnaires asked them about their experience regarding the ways in which the issues of race and culture affect counseling, psychotherapy and psychological intervention. The other [50] tried to offer a dynamic point of view about the relationship between a mental health professional and a migrant patient by considering the construct of counter-transference.

From our analysis, we found that the presence of judgment biases and distortions in such a situation was controversial. In fact, four of the studies [11,44,52,53] did not provide evidence of any variation in variables such as diagnosis, global evaluation or the quality of the therapeutic relationship that was influenced by cultural variables. This result was interpreted as evidence of the absence of biases and prejudices toward migrant patients. However, the use of clinical vignettes assumed that the diagnosis was an individual cognitive process that could be isolated from the context, with ethnicity deemed an independent variable [53]. Hence, we did not consider the possible contingencies that can play an influential role in a real clinical situation. Therefore, one must remain cautious in interpreting these results.

## 3. Results

All the studies highlighted the presence of biases that could be classified into three macro-categories: the activation of racial stereotypes, ethnocentrism and micro-aggressions. Each of these three categories is discussed below.

### 3.1. Racial Stereotypes 

Some of the selected studies highlighted how stereotypes and cultural prejudices can affect the therapeutic pathway. These judgmental errors are based on the so-called representativeness heuristic [69], which leads to the categorization of one stimulus based on its similarity with another stimulus or a set of stimuli. This mental operation results in an improper generalization of judgments and evaluations used within a specific set of stimuli to easily categorize a new stimulus.

As per our review, an ethnic minority patient might be seen as a member of the category “Afro-American”, “Hispanic”, or simply “different” or “other”. Subsequently, mental health professionals may use characteristics and peculiarities attributed to a specific category throughout the process of knowing their patients.

The contents of these prejudices, as individuated by the studies, can be organized into three categories. The first type of prejudice regards the dangerousness and the violent nature of patients belonging to specific ethnic groups, particularly Afro-Americans. It was found that this type of prejudice may lead mental health professionals to use tranquilizers, isolation and restraints in a more consistent way and decide not to use recreational and occupational therapies [14]. According to a study by Lawson, Hepler, Holladay and Cuffel [54], this kind of cognitive bias may be one of the reasons behind the higher number of hospitalizations among black mental patients compared with Caucasian mental patients. Such a type of prejudice may also influence a diagnosis: Afro-American patients were found to be more likely to receive a diagnosis consistent with violence issues, aggressiveness and suspiciousness compared with patients of other ethnicities [21,51].

The second type of prejudice concerns work issues. Jones and Grey’s [15] study found that most mental health professionals consider the symptoms shown by black patients as issues concerning their working situation and tend to exclude the possibility of mood disorders. Three other studies [58,60,61] found that in a counseling setting, culture might affect the evaluation of a patient’s future work-related potential. The authors interpreted these results as evidence for the argument that among mental health professionals, stereotypes involved in evaluating the clinical situation of patients may lead to divergent conclusions.

The third type of prejudice concerns the abuse of alcohol by Afro-American people. The study by Luepnitz, Randolph and Gutsh [48] showed that in a group of white and black patients with the same symptoms, the latter were more likely to receive a diagnosis of alcoholism owing to the widespread common sense that black people drink more than white people.

The fourth and fifth types of prejudice could possibly lead mental health professionals to evaluate black and white patients’ origins in different ways based on some outcomes from the scientific literature: black patients were assessed to have lower levels of verbal skills [12] and a higher rate of schizophrenia [47] compared with white people. In the study by Bell and Mehta [47], a phenomenon called misdiagnosis emerged: most black patients were more likely to receive an improper diagnosis of schizophrenia, even if their symptoms were more consistent with a diagnosis of manic depressive disorder. In fact, many studies underlined the issue of the overdiagnosis of schizophrenia [17,20,49,59].

Related to stereotypical representations intended as a strong and rigid opinion not acquired by experience and independent of a single case’s evaluation, we need to consider a mental health professionals’ possible perception that cultural differences make it difficult to manage current or future clinical situations [14,15,42,46]. This kind of judgmental error can be seen as a *prevision*, and such a mental operation assumes that an empirical bond exists between two entities [70]: considering the ontological existence of the entity-cause (in our case, cultural diversity), it is possible to foresee the entity-effect (in our case, difficulty and stress). In our review, this prevision influenced how mental health professionals presented themselves to and acted toward migrants [71]; the clinicians evaluated the cross-cultural clinical situation as more challenging and harder even before meeting their patients.

A common issue in several of the selected studies concerned these prejudices’ implicit or explicit nature. Three of the studies [57,64,66] used different methods (presentation of prime words, Implicit Association Test) to investigate implicit stereotypes toward immigrant patients. Two of these three studies [57,64] provided evidence of implicit biases. Katz and Hoyt [66] found that the variance of an outcome expectations scale among clinicians was fully explained by the score concerning the explicit nature of their prejudices. Nevertheless, they suggested that social desirability may be a confounding variable among mental health professionals. 

### 3.2. Ethnocentrism

Another error emerging from the studies was ethnocentrism. This word refers to the interpretation of a behavior or a sentence acted or pronounced by the patient and detected by the mental health professional based on psychological theories and practices derived from the standards and attitudes normalized by the professional’s native culture. As per our review, a judgmental error assumes that the cognitive categories, behaviors and symptoms shown by the patients are the same as those associated with Western cultural scripts. Indeed, in an illustrative research, Li-Repac [45] showed that mental health professionals tended to evaluate Asian patients as more depressed and inhibited than their Caucasian counterparts. According to the researcher, these differences could stem from a “Western” interpretation of the behavior of Chinese clients. In fact, “the value placed on being frank and open by American culture contrasts with the Chinese tendency to be quiet, to listen to, and to be cautious about one’s effect on others” [65, p. 338]. Thus, the difference in interpersonal style might be seen by a white therapist as a sign of social introversion and even diagnosed as depression.

Similarly, in another study, Arroyo [55] underlined that differences in pronunciation and accent in Hispanic patients were interpreted as signs of reduced emotional expressiveness or blunted affect. In this study, the cultural background influenced the interpretation of patients’ speech; thus, linguistic differences were confused with emotional withdrawal.

Two other studies [28,62] focused on ethnocentrism regarding assessment instruments: specific answers in diagnostic interviews were immediately linked to a certain diagnosis. However, the diagnostic system’s norms did not adequately consider variations among different cultures. Thus, this phenomenon led mental health professionals to describe the patients in front of them with labels that were deemed as lacking validity.

Ridley [50] tried to provide a psychoanalytic point of view to the above question by proposing the paradigm of *pseudo-transference*, a phenomenon reportedly occurring during interracial therapy. According to the author, some defensive reactions of black clients were triggered by behaviors and stereotyped attitudes among white therapists. Consequently, the therapist might have misinterpreted and labeled their patient’s response as pathological even though the reaction was justified. 

### 3.3. Microaggression

Another form of bias that emerged from our systematic review concerned microaggressions. This type of bias includes a wide range of judgmental errors that, in some cases, could be classified as belonging to the categories we proposed above. However, because of the focus received by this error and its peculiar form, it was necessary to diversify this category from those previously examined. In 1978, the first author to use the term *microaggression* was Chester Pierce [72], who defined it as a subtle, stunning, often automatic and non-verbal exchange that berates a person. In 2007, Sue et al. [73] (p. 273) reused this term by specifying its meaning: “brief, everyday exchanges that send denigrating messages to people of color because they belong to a racial minority group”. In any case, microaggressions consist of a multiplicity of attitudes and communications, intentional and unintentional, which depict a lack of sensibility, respect and attention toward some aspects of a culture that is different from someone’s own. 

Sue et al. [73] described three different types of microaggressions. First, *Microassaults* are severe offenses, always explicit and intentional, involving denigrations of an individual’s racial group (e.g., referring to someone as “colored”). Second, *Microinsults* are subtler and more unconscious communications that put down an individual’s racial group (e.g., asking a person of color, “how did you get this job?”). Third, *Microinvalidations* are communications that tend to negate or deny the thoughts, feelings or experiences of a person of color (e.g., telling a person of color, “I don’t see color”). Sue et al. [73] showed that the last two are quite common in the field of counseling.

Further, Constantine [63] identified 12 categories of racial microaggressions that can occur in a counseling context: (a) colorblindness, (b) overidentification, (c) denial or personal or individual racism, (d) minimization of racial-cultural issues, (e) assignment of a unique or special status based on race or ethnicity, (f) stereotypical assumptions about members of a racial or ethnic group, (g) accused hypersensitivity regarding racial or cultural issues, (h) meritocracy myth, (i) culturally insensitive treatment considerations or recommendations, (j) acceptance of less-than-optimal behaviors based on racial or cultural group membership, (k) idealization and (l) dysfunctional helping/patronization. These categories were converted into a 12-item, three-point Likert-type questionnaire, the Racial Microaggressions in Counseling Scale (RMCS), measuring respondents’ perceptions of racial microaggressions in counseling and the perceived impact of these microaggressions on them. 

According to researchers, many of the racial microaggressions committed in counseling rooms have less to do with the counselor saying or doing something offensive than with minimizing the importance of cultural issues or communicating defensiveness or discomfort with regard to being reminded about one’s biases or prejudices [67,74]. Moreover, the working alliance appeared to moderate the impact of perceived microaggressions on clients’ psychological well-being [65]. Clients who perceived microaggressions with their therapists but maintained a high alliance still reported improvements: a strong alliance could temper the negative impact of ruptures caused by microaggressions. 

The nature of microaggressions as subtle and unconscious phenomena could prevent therapists from recognizing that their clients experience such offenses. Furthermore, therapists are deemed reluctant and uncomfortable when addressing issues of race and ethnicity. In fact, client and therapist dyads that discussed the microaggression experiences had higher-quality alliances than those that did not discuss it, being deemed similar to the ones in which there was no perceived microaggression. According to some researchers, this result revealed the benefit of addressing the missteps that can occur during therapy [67].

Hook and colleagues [68] aimed to examine how the perception of cultural humility is associated with the frequency and impact of microaggressions. The researchers defined cultural humility as “the ability to maintain an interpersonal stance that is other-oriented (or open to the other) in relation to aspects of cultural identity that are most important to the client” [68] (p. 2). Higher rates of perceived cultural humility in therapists were found to be associated with lowered frequency and impact of racial microaggressions. These findings were consistent with the hypothesis concerning the importance of this characteristic during therapy involving ethnically diverse clients. Counselors who were perceived by their patients as having high levels of cultural humility were deemed less likely to commit racial microaggressions; when they did so, they were able to acknowledge and admit their limitations and mistakes regarding cultural issues.

## 4. Discussion 

This review was aimed at evaluating the presence and typology of cognitive errors committed by mental health professionals toward those patients who are perceived as culturally different. Although some of the studies denied the existence of these errors, most of them contributed to confirming the hypothesis regarding the presence of cognitive mistakes and heuristics in mental health services; this mistake could be divided into three main categories. 

The first category concerns stereotypes. In the clinical encounter with the immigrant patient, mental health professionals can be influenced by the information concerning a patient’s cultural background. This information can contribute to the formulation of pre-judgments based on ideas and beliefs that are commonly accepted or theories that belong to the observers’ native cultural scripts, along with notions of common sense, none of which are valid in a scientific way. The results of our review also showed that stereotypes could be divided into five different areas:-Violent nature, which concerns the unjustified attribution of characteristics to a supposed violent nature of the patient. This attribution leads to cognitive distortions concerning diagnostic evaluation and treatment indications;-Work issues that deem the consideration of problems in the patient’s work-related sphere as central to diagnostic evaluation and related to their future potential;-Alcohol abuse, which concerns the belief regarding the diagnostic evaluation that problems concerning the abuse of alcoholic substances are diffused and common among people of specific ethnicities;-Lower verbal skills, which concern a prejudicial negative evaluation regarding the patient’s verbal capacity that leads to distortion during diagnostic and treatment processes;-Higher rates of schizophrenia, which concern the act of basing an individual diagnosis on the epidemiological frequency of a diagnosis within a specific ethnic group; this leads to the phenomenon of misdiagnosis.

A further critical aspect regarding the judgmental errors related to stereotyping concerns foreseeing a higher degree of difficulty and stress in the management of culturally distant patients. This prediction is epistemologically unfounded since it is based on applying a modality of knowledge that is typical of a mechanistic paradigm. The latter postulates the existence of empirical relations within ontological entities [75]. This error may have significant effects on the interaction between therapists and patients during the clinical pathway regarding diagnostic judgments and evaluations of the displayed symptoms.

The second category of errors concerns the phenomenon of ethnocentrism. In this case, mental health professionals take for granted that the cognitive categories they use to define their experiences are identical to the ones used by their patients. This error in the interpretation of some behaviors shown by the patients and in the use of assessment instruments that are valid for the Western cultural scripts may have substantial consequences for the culturally different patients because their behaviors are classified using diagnostic labels that lack validity, as they do not account for cultural variations.

The third category of judgmental error concerns the phenomenon of microaggressions, a subtle bias appearing in the communicative exchange between professionals and immigrant patients. According to the studies we reviewed, in clinical situations, verbal and non-verbal exchanges often contain offensive messages toward patients whose cultural backgrounds are different from those of mental health professionals. Moreover, microaggression can involve mental health professionals using interactive modalities that minimize the emphasis on cultural issues and expressing discomfort at becoming aware of bias, stereotypes and prejudices shown toward some patients. Thus, microaggressions affect the perceived therapeutic alliance and can provoke therapy drop-outs.

It is essential to underline that the errors found in our analysis of the literature may be both implicit and explicit. Indeed, several of the selected studies, characterized by specific methodologies, emphasized that the reasons leading to decisional processes (diagnostic evaluation, symptoms interpretations and treatment) could be unconscious.

### Limitations and Future Research

First, this review is limited in its concerns over the scarcity of specific studies involving patients from different cultural backgrounds. It focuses on research conducted in the Western academic world; when discussing ethnic minorities or migrants from unfamiliar areas, there is, thus, the risk of taking many concepts for granted. It is important to underline that most of the studies we examined were conducted in the USA; some of its ethnic minorities are indigenous. This factor may have guided the results and analyses, and therefore, one must tread with caution in generalizing the results. Furthermore, there was considerable variability among the studies included, such as the number of participants, survey methods and evaluation of the interventions. Future studies should aim to investigate the topic under review in different contexts, e.g., the European context. 

## 5. Conclusions

According to the World Bank and ONU data, it is possible to anticipate that in the next 35 years, the total population of the world will increase by 50%, especially in the poorest parts of the world, in addition to a population decrease in most European countries. This would probably intensify migration flows from the poorest and most populous parts of the world to the richest and least populous areas. In this scenario, reflections regarding the interactive modalities aimed at welcoming communities and migrants appear to be imperative. Even in the psychological and clinical field, it is vital to confront this theme since the number of migrants who seek help will increase not only in public health services but also in private facilities. Results from the available studies clarify that during the clinical encounter and the interpretative process, mental health professionals are likely to resort to biases and distortions. The clinical encounter with migrant patients is experienced as a problem to which the majority of mental health professionals respond by adopting heuristics in order to reduce the feeling of disorientation. Instead, it is crucial to adopt an intercultural perspective according to which the culturally different person is not seen as “other” or “diverse” but as an actor in a co-participated process of community building. That is why it is particularly important to help mental health professionals recognize their own biases and cognitive issues and to put them in interaction with those of the users in a co-participatory process that can renew the identity of psychological work and the mental health service itself. Only in this way can alterity be valued and perhaps self-knowledge be continued over time.

## Figures and Tables

**Figure 1 behavsci-14-00155-f001:**
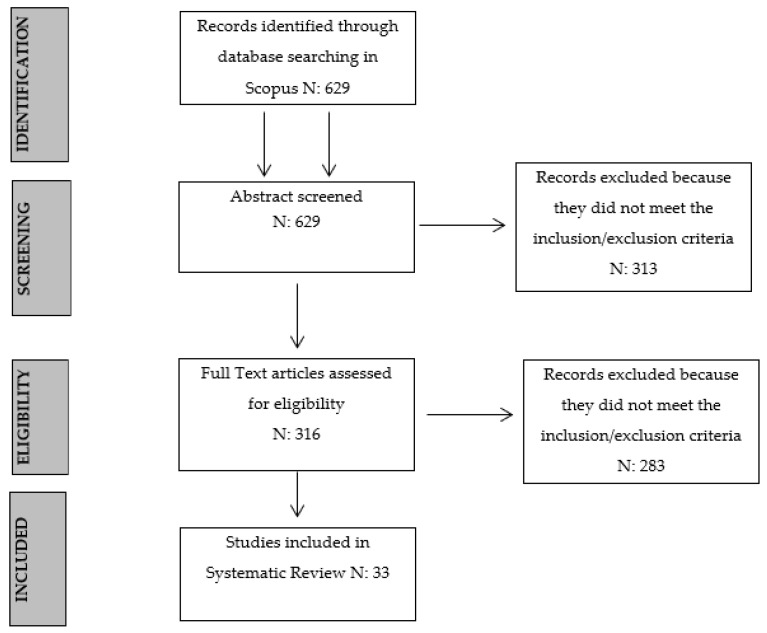
PRISMA flow diagram illustrating the processes of literature searches and screening.

**Table 1 behavsci-14-00155-t001:** Selection of studies.

	Title	Author	Country	Purpose of Study	Type of Study and Method(s)
1	Racial attribution effects on clinical judgment: A failure to replicate among white clinicians	Bloch, Weitz, Abramowitz 1980 [44]	USA	To investigate the variation in the countertransference phenomenon according to the patient’s ethnicity	Research Article; Analyses of Variance
2	The effects of race and social class on clinical judgement	Bamgbose, Edwards 1980 [11]	USA	To identify the presence of racial and/or social bias in clinical judgement	Research Article: Correlational Research
3	Cultural influences on clinical perception	Li-Repac 1980 [45]	USA	To investigate the degree to which clinical perceptions are affected by the cultural distance between therapist and client	Research Article; Correlational Research
4	Measuring racial bias in inpatient treatment	Flaherty, Meagher 1980 [14]	USA	To investigate the presence of racial bias in a psychiatric hospitalization unit	Research Article; Analyses of Variance
5	Cross-racial psychotherapy: what the therapists say	Turner, Armstrong 1981 [46]	USA	To investigate the experience of cross-cultural psychotherapy among therapists	Research Article; Analyses of Variance
6	Misdiagnosis of black patient with manic depressive illness: second in a series	Bell, Mehta 1981 [47]	USA	To investigate the phenomenon of misdiagnosis of manic depressive illness in black patients	Case Report
7	Race and socioeconomic status as confounding variables in the accurate diagnosis of alcoholism	Luepnitz, Randolph, Gutsch 1982 [48]	USA	To determine whether bias and/or prejudices related to race or socioeconomic status complicate the diagnosis of alcoholism	Research Article; Analysis of Variance
8	Misdiagnosis of schizophrenia in bipolar patients: a multiethnic comparison	Mukherjee, Shukla, Woodle, Rosen, Olarte 1983 [49]	USA	To assess whether there is a significant correlation between ethnicity and misdiagnosis	Research Article; Correlational Research and Univariate Analysis
9	Black and white psychiatrists: therapy with blacks	Jones, Gray 1985 [15]	USA	To investigate the therapist’s experience of cross-cultural psychotherapy	Research Article; Correlational Research and Univariate Analysis
10	Pseudo-transference in interracial psychotherapy: an operant paradigm	Ridley 1985 [50]	USA	To present a paradigm regarding the phenomenon of pseudo-transference in cross-cultural psychotherapy	Theoretical Article
11	Diagnostic judgements as a function of client and therapist race	Strickland, Jenkins, Myers, Adams 1988 [12]	USA	To examine the client’s evaluation process according to the ethnicity of the client and the therapist	Research Article; Multivariate Analysis
12	Gender, Race, and DSM—III: A study of the objectivity of psychiatric diagnostic behavior	Loring, Powell 1988 [18]	USA	To investigate whether psychiatrists provide diagnosis based on gender and ethnicity using the third edition of the Diagnostic Manual of Mental Disorders (DSM-III)	Analogue Study; Univariate and Multivariate Analysis
13	Psychiatric diagnoses and racial bias: an empirical investigation	Pavkov, Lewis, Lyons 1989 [20]	USA	To investigate to what extent being black is linked to a diagnosis of schizophrenia, independent of other clinical and social variables	Research Article; Logistic Regression
14	Are British psychiatrists racist?	Lewis, Croft-Jeffreys, David 1990 [51]	UK	To investigate racial stereotypes among British psychiatrists	Research Article; Univariate and Multivariate Analysis
15	Racial/ethnic identity and amount and type of psychiatric treatment	Flaskerud, Hu 1992 [52]	USA	To examine the relationship between ethnic identity and type of psychiatric treatment	Research Article; Multiple Regression Analysis
16	Psychiatric diagnosis and racial bias: empirical and interpretative approaches	Littlewood 1992 [53]	UK	To investigate the reasons for the increase in the number of mental disorders among black people in England	Theoretical Article and Case Study
17	Race as a factor in inpatient and outpatient admission and diagnosis	Lawson, Hepler, Holladay, Cuffel 1994 [54]	USA	To determine whether African American Patients are over-represented in the use of a certain type of treatment setting and diagnostic categories	Research Article; Correlational Study
18	Psychotherapist bias with Hispanics: an analog study	Arroyo 1996 [55]	USA	To examine the influence of ethnicity on the clinical judgment of psychotherapists	Analogue Study; Multivariate Analysis
19	The experienced influence or effect of cultural/racism issues on the practice of counseling psychology—a qualitative study of one multicultural training organization	Clarkson, Nippoda 1997 [56]	UK	Examine how issues related to race and culture affect the practice of counseling, psychology, and psychotherapy	Qualitative research; Discourse Analysis
20	Ethnicity/race, paranoia, and psychiatric diagnoses: clinician bias versus sociocultural differences	Whaley 1997 [28]	USA	To conduct a comparative study between two different hypotheses to explain paranoid symptoms	Research Article; Univariate and Multivariate Analysis
21	Conscious and Nonconscious african american stereotypes: impact of first impression and diagnostic ratings by therapist	Abreu 1999 [57]	USA	To evaluate the presence of racial bias using a priming stimulus	Research Article; Priming Procedure and Multivariate Analysis
22	Effects of client race on clinical judgement	Rosenthal, Berven 1999 [58]	USA	To investigate the influence of racial bias on the counselor’s clinical judgment	Research Article; Univariate Analysis and Correlational Research
23	Clinician attributions associated with the diagnosis of schizophrenia in African American and non-African American patients	Trierweiler, Neighbors, Munday, Thompson, Binion 2000 [17]	USA	To investigate whether clinicians diagnose schizophrenia in African American and American patients differently	Research Article; Multivariate Analysis and Logistic Regression
24	Racial differences in DSM Diagnosis Using a Semi-structured instrument: the importance of clinical judgment in the diagnosis of African Americans	Neighbors, Trierweiler, Ford, Muroff 2003 [59]	USA	To investigate the relationship between the patient’s ethnicity and the diagnostic process	Research Article; Correlational Research
25	Effects of client rate on clinical judgement of african american undergraduate students in rehabilitation	Rahimi, Rosenthal, Chan 2003 [60]	USA	To investigate the influence of racial bias on the counselor’s clinical judgment	Research Article; Univariate and Multivariate Analysis
26	Effects of client race on clinical judgement of practicing european american vocational rehabilitation counselor	Rosenthal 2004 [61]	USA	To investigate the influence of racial bias on the counselor’s clinical judgment	Research Article; Multivariate Analysis and Logistic Regression
27	Ethnicity and mental health treatment utilization by patients with personality disorders	Bender, Skodol 2007 [62]	USA	To examine the relationship between ethnicity and treatment in patients diagnosed with personality disorder	Research Article; Univariate Analysis and Correlational Research
28	Racial microaggressions against African American clients in cross-racial counseling relationships	Constantine 2007 [63]	USA	To examine the relationship between African American clients and Euro-American counselors	Research Article; Implicit Association Test and Correlational Research
29	The relationship between level of training, implicit bias, and multicultural competency among counselor Trainees	Boysen, Vogel 2008 [64]	USA	To investigate cultural competencies and implicit biases among counselors	Research Article; Univariate Analysis and Correlational Research
30	Cultural ruptures in short-term therapy: working alliance as a mediator between clients perceptions of microaggressions and therapy	Owen, Imel, Tao, Wampold, Smith, Rodolfa 2011 [65]	USA	To determine if the client’s perception of microaggression varies according to the ethnicity	Research Article; Univariate and Multivariate Analysis
31	The influence of multicultural counseling competence and anti-black prejudice on therapists outcome expectancies	Katz, Hoyt 2014 [66]	USA	To examine the effect of biases and multicultural skills on the therapist’s expectations of results	Research Article; Univariate and Multivariate Analysis
32	Adressing racial and ethnic microaggressions in therapy	Owen, Tao, Imel, Wampold, Rodolfa 2014 [67]	USA	To investigate the effect of microaggressions in therapy	Research Article; Univariate and Multivariate Analysis
33	Cultural humility and racial microaggression in counseling	Hook, Farrell, Davis, DeBlaere, Van Tongeren, Utsey 2016 [68]	USA	To investigate the frequency and the impact of microaggression in counseling	Research Article; Univariate and Regression Analysis

## Data Availability

The dataset presented is available under request.

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
