# Peer review of "The Invisible Discrimination: Biases in the Clinical Approach Regarding Migrants: A Study to Help Ethnopsychology Services and Clinicians"

_behavsci, 2024, doi:10.3390/bs14030155_

Round 1
Reviewer 1 Report
Comments and Suggestions for Authors
Review on "The invisible discrimination: biases in the clinical approach with migrants. A study to help Ethnopsychology Services and clinicians"
Thank you for submitting this interesting and important paper. The significance of the topic resonates both with academics and with practitioners in democratic countries. However, I'd like to address a few concerns to enhance the overall coherence and clarity of your work.
The primary issue revolves around the consistency of the Research Question (RQ) throughout the paper. Although the authors pose the RQ as "How do the country of origin and the related culture affect the decision-making process of mental health professionals in a clinical encounter?" in the introduction, this focus seems to drift in subsequent sections. Specifically, the paper discusses stereotypes but lacks a clear connection to their impact on decision-making or therapy model selection. It is crucial to rephrase the RQ to better align with the paper's content and ensure a cohesive narrative.
Moreover You should connect the next coming passage to the RQ . It seems now disconnected. :“In this case, we are interested in the use and the meaning of the word “culture” which can be misleading for mental health professionals, who may be more likely to commit errors in the understanding of their patients when this word is used as a “matter of fact”. We think that a non-reified conception of culture might help to overcome the impasses which lead patients and professionals to misunderstand each other’s communications and behaviors. More about this topic will be explored in the Conclusion paragraph.”
For instance, the abstract mentions errors related to stereotypes, yet the findings diverge from this theme. Additionally, when posing the RQ, it is essential to specify what aspect of decision-making or clinical encounter it pertains to. Subsequent sections should be seamlessly connected to the RQ, avoiding any sense of disjointedness.
In the Discussion section, the literature review introduces cognitive errors committed by mental health professionals towards culturally different patients. However, this content does not adequately tie back to the study's aim or RQ. Establishing a more substantial connection between these elements is necessary.
Furthermore, the Conclusion presents a different objective from what was initially stated in the study. Clarifying this discrepancy is essential to maintain coherence.
Other noteworthy points for improvement include language editing in the abstract and providing a more detailed explanation in the Introduction. Elaborate on how the summarized literature relates to mental care, distinguishing factors affecting both patients and staff.
The next session needs more elaboration. It seems like it summarizes the literature but unfortunately the reader cannot understand the added value of this summary. For example how these aspects affected the mental care? The authors should divide this session into factors related to patients and to the staff.
“The most convincing causes found about the disparities concerning the diagnosis were: cultural differences in symptomatology [17], poor usage or inequalities in public mental health services [18, 19], distrust toward western therapy [20], missing validity of tests and assessment instrument [21, 22], and judgmental errors committed by the mental 60 health professionals [23-25], and bullyism [26]”.
In the Methods section, consider providing a precise definition of "immigrant" and elaborate on the limitations of the methodology. Connect the findings more explicitly to the study's aim and RQ to enhance the overall alignment of your work.
Please provide more Limitation of this specific methodology
it seems that this method did not provide the answer on your RQ. Please connect the findings to the aim of your study and RQ.
Reviewer 2 Report
Comments and Suggestions for Authors
This is a veyr interesting study exploring biases in the clinical practice among migrant individuals. The main goal of the paper was to identify frequent errors or difficulties in treating these patients in order to help clinicians to improve their practices.
The paper is well-written, and of interest for the readers; however, several minor changes are recommended before considering it for publication.
ABSTRACT
1- I recommend to summarize the first part that introduces the topic of migration and culture.
2- The main purpose is to summarize the evidence about errors or prejudices ocurring in clinical practices among people from different cultural background? Please, reformulate it more concise.
2- Three main biases were identified: racional stereotype activation, ethnocentrism and micro-aggression. How many papers were found for each of them?
INTRODUCTION
1- I recommend to divide the target population into three (almost) subpopulations: migrants, migrant refugees, and rural-to-urban migrant workers. These three groups can be different in terms of needs and expectatives.
METHODS
1- Why did the authors use the Scopus database and not PubMed?
2- Please, divide the methods section into several subtypes according to the screening and selection process, search terms, inclusion and exclusion criteria, etc.
3- Is it really a systematic review?
RESULTS
1- Section 3.2. is followed by 3.4. Please, renumber them.
Have the authors identified any other bias? Did they find only these three or were they the most representative?
CONCLUSIONS
The conclusions section should not be just a summary of the findings, and should not include references.
